# Sex-specific fear acquisition following early life stress is linked to amygdala and hippocampal purine and glutamate metabolism
Joeri Bordes [1], Thomas Bajaj[2], Lucas Miranda[3,4], Lotte van Doeselaar[1,4], Lea Maria Brix[1,4], Sowmya Narayan[1,4], Huanqing Yang[1], Shiladitya Mitra[1], Veronika Kovarova[1,4], Margherita Springer[1], Karin Kleigrewe [5], Bertram Müller-Myhsok [3], Nils C. Gassen [2] & Mathias V. Schmidt [1] ✉

Early life stress (ELS) can negatively impact health, increasing the risk of stress-related disorders, such as post-traumatic stress disorder (PTSD). Importantly, PTSD disproportionately affects women, emphasizing the critical need to explore how sex differences influence the genetic and metabolic neurobiological pathways underlying trauma-related behaviors. This study uses the limited bedding and nesting (LBN) paradigm to model ELS and investigate its sex-specific effects on fear memory formation. Employing innovative unsupervised behavioral classification, the current study reveals distinct behavioral patterns associated with fear acquisition and retrieval in male and female mice following ELS. Females exposed to LBN display heightened active fear responses, contrasting with males. Furthermore, the study examined the crucial link between behavioral regulation and cellular metabolism in key brain regions involved in fear and stress processing. Sex-specific and stress-dependent alterations were observed in purine, pyrimidine, and glutamate metabolism within the basolateral amygdala, the dorsal hippocampus, and the ventral hippocampus. These findings provide crucial insights into the complex interplay between metabolic pathways, the neurobiological underpinnings of fear memory, and stress responses. Importantly, they emphasize the significance of considering sex-specific metabolic alterations when investigating stress-related disorders, opening potential avenues for the development of targeted interventions.

Early life stress (ELS) exposure, such as child abuse or neglect, has severe, long-lasting negative behavioral and physiological consequences in adulthood, including, among others, an altered neuroendocrine function[1–3] and morphological changes in the brain[4,5]. This ultimately leads to an increased risk and persistence of stress-related disorders, such as post-traumatic stress disorder (PTSD)[6–8]. Human genetic studies have identified that stress-related disorders are partially mediated by different genomic variations[9–15], in particular the glucocorticoid receptor (GR) co-chaperone FK506 binding protein 51 kDa (FKBP51), encoded by the *FKBP5* gene[16–19]. An additional

important factor influencing the susceptibility and severity of PTSD is sex. The prevalence of PTSD is around twice as high in women (11.0%) compared to men (5.4%)[20]. However, the increased risk for PTSD development in women could not be entirely explained by differences in the event type or severity of the traumatic event, which might indicate a potential underlying biological mechanism[21,22]. Therefore, investigating the underlying mechanisms related to ELS exposure in a sex-specific way is crucial to advance the understanding of the neurobiological mechanism of PTSD.

[1]Research Group Neurobiology of Stress Resilience, Max Planck Institute of Psychiatry, 80804 Munich, Germany. [2]Neurohomeostasis Research Group, Department of Psychiatry and Psychotherapy, Bonn Clinical Center, University of Bonn, 53127 Bonn, Germany. [3]Research Group Statistical Genetics, Max Planck Institute of Psychiatry, 80804 Munich, Germany. [4]International Max Planck Research School for Translational Psychiatry (IMPRS-TP), 80804 Munich, Germany. [5]Bavarian Center for Biomolecular Mass Spectrometry, TUM School of Life Sciences, Technical University of Munich, Freising, Germany. ✉e-mail: mschmidt@psych.mpg.de

An increasing number of clinical studies point to corticolimbic structures as strongly affected in size and functioning by ELS exposure, in particular the basolateral amygdala (BLA) and hippocampus (HIP)[23–27]. Moreover, the BLA and HIP continue their development in function and morphology during the early postnatal period[28–30], rendering this developmental period especially vulnerable to environmental insults. Furthermore, both the BLA and HIP exhibit high expression levels of GRs and mineralocorticoid receptors (MRs)[31–34], making them particularly vulnerable to the effects of ELS exposure. Animal models have further elucidated a significant role of FKBP5 in these regions, demonstrating a high expression under baseline conditions, and a strong upregulation following stress[35].

Alterations of anxiety and fear behavior are a central hallmark of a PTSD-like phenotype in animal models. A study using a mouse model, overexpressing the human FKBP5 gene in the forebrain showed that elevated FKBP5 expression in combination with ELS exposure increases anxiety-related behavior, which was more pronounced in females[36]. Therefore, the genetic regulation of FKBP5 is linked to affect males and females differently after the exposure to ELS[36–38], but the impact of ELS exposure on FKBP5 regulation remains to be investigated. There is an increasing body of evidence that ELS exposure affects rodents in a sex-specific manner[39–43]. Exposure to fear conditioning in rodents has been shown to strongly activate the BLA and HIP in a time-dependent manner[44,45]. The important role of the BLA and HIP in ELS exposure indicates that the formation of fear acquisition and memory could be affected by ELS exposure. Previous research has shown that ELS reduces freezing responses during contextual as well as auditory fear memory retrieval in males, which was linked to a reduction of synaptic plasticity markers in the dorsal HIP[46]. Furthermore, recent work revealed sex-specific effects of ELS during fear acquisition, with ELS males displaying reduced freezing responses compared to nonstressed conspecifics, a pattern not observed in females[47]. However, the exact neurobiological mechanism and behavioral strategies underlying ELS-induced alterations of fear memory formation between sexes remains to be uncovered. The exploration of metabolic mechanisms and pathways (metabolomics) in the brain is an important assay, which enables the analysis into the final products of cellular processes. Consequently, metabolomics can reflect potential disease states and has been utilized as a tool for biomarker discovery[48]. Therefore, a metabolic analysis approach can contribute to the understanding of the underlying neurobiological mechanisms related to the sex-dependent effects of ELS exposure. Previous research has shown the importance of the metabolic purine and pyrimidine pathways in the HIP when looking at successful treatment response of chronic intervention with the selective serotonin reuptake inhibitor paroxetine[49]. However, the exploration of metabolism and the underlying regulatory pathways in specific stress-related and affected brain regions in relation to ELS has remained uninvestigated.

In the current study, we address these open questions by investigating the sex-specific effects of ELS exposure using the established limited bedding and nesting (LBN) paradigm on FKBP5 expression, brain tissue metabolomics, and fear acquisition and retrieval. Utilizing an unsupervised deep phenotyping strategy, we show that specific aspects of fear behavior and memory are altered by ELS in a sex-dependent manner. Moreover, metabolic pathway analysis in the BLA, dorsal HIP (dHIP) and ventral HIP (vHIP) was performed and identified, for the first time, altered purine metabolism in the BLA, dHIP, and vHIP between sexes, while an interaction effect of sex and LBN exposure was observed for glutamate metabolism specifically in the BLA. The present study highlights the intricate interplay between stress-responsive genes, metabolic pathways, and the neurobiological substrates implicated in fear memory formation and stress regulation.

## Materials and methods
### Animals
Adult male and female C57/Bl6N mice (age between 2-3 months of age) were obtained from the in-house facility of the Max Planck Institute of Psychiatry and used for breeding ($F_0$). Animals from the $F_1$ generation were used as experimental animals and were weaned at P25 in groups of maximum four animals with their littermates. Animals were housed in individually-ventilated cages (IVC; 30 cm × 16 cm × 16 cm connected by a central airflow system: Tecniplast, IVC Green Line—GM500). All animals were kept under standard housing conditions; 12 h/12 h light-dark cycle (lights on at 7 a.m.), temperature 23 ± 1 °C, humidity 55%. Food (Altromin 1324, Altromin GmbH, Germany) and tap water were available ad libitum. All experimental procedures were approved by the committee for the Care and Use of Laboratory Animals of the government of Upper Bavaria, Germany. All experiments were in accordance with the European Communities Council Directive 2010/63/EU.

### Early life stress paradigm: limited bedding and nesting
Early life stress (ELS) was performed using the limited bedding and nesting (LBN) paradigm to induce chronic stress towards the mother and pups during P02 to P09, as previously described by Rice et al.[50]. See supplemental methods for details.

### Adult behavioral testing
At 3 months of age, a cohort of both males and females, were tested on a fear conditioning protocol containing fear acquisition (day 1) and the subsequent recall of contextual fear memory (day 2) and auditory fear memory (day 3). The behavioral tests were performed between 8 a.m. and 11 a.m. in the same room as the housing facility.

### Fear conditioning
A 5-time conditioned-unconditioned stimulus pairing was utilized as a fear conditioning protocol to maximize the fear behavior read-outs, by combining a contextual and cued fear conditioning model[51]. The fear conditioning protocol was performed as previously described (see Supplemental Methods for details)[52,53]. Data were recorded and analyzed using the ANY-maze 7.2 software (Stoelting, Ireland), in which the percentage of the time freezing was calculated. Furthermore, the fear acquisition data were subsequently analyzed using DeepLabCut version 2.2b7[54] and DeepOpenField (DeepOF) version 0.2[55,56] for the unsupervised analysis pipeline.

### Unsupervised analysis of the fear conditioning data
An additional unsupervised analysis was performed for the fear acquisition data in both males and females in order to obtain a more in-depth analysis of the behavioral differences between conditions and sexes during the ITIs and tones 2–5. First, pose estimation was performed on the raw data videos using DeepLabCut version 2.2b7 (single animal mode). DeepLabCut pose estimation analysis was performed using 11 body parts, including the nose, left and right ears, three points along the spine (including the center of the animal), all four extremities, and the tail base.

Subsequently, DeepLabCut annotated datasets were processed and analyzed using DeepOF v0.2, as described previously[55,56]. In brief, DeepOF preprocesses the DeepLabCut annotated data by performing rotational alignment and centering of the coordinates. The unsupervised analysis of the fear acquisition data was performed on the entire video length and utilized the same model (DeepOF - Variational Deep Embeddings (VaDE)) for the male and female data in order to make cluster interpretation between sexes possible. The interpretation of the clusters was explored by visual inspection of representative video snippets for each specific cluster. Representatives were selected as instances with a cluster assignment confidence greater or equal than 0.9. In addition, Shapley additive explanations (SHAP) were utilized to rank feature importance per cluster in order to further understand the expressed behavior within clusters.

### In-situ hybridization of FKBP5
The FKBP5 mRNA profile was determined using radio-active in-situ hybridization labeling as described previously[35]. See supplemental materials for details.

## Metabolomics

**Tissue collection**. Mice were decapitated, and their brains were immediately removed and snap-frozen using 2-methylbutane (cooled on dry ice) and stored at −80 °C until further use. Brains were sectioned into 250 μm coronal slices using a cryostat, targeting specific regions: BLA (6 slices, bregma: −0.70 to −1.91), dHIP (3 slices, bregma: −1.34 to −2.09), and vHIP (3 slices, bregma: −2.80 to −3.52). Bilateral punches were performed on the slices using a sample corer (Fine Science Tools, Item No. 18035-01), and the tissue was stored at −80 °C until further analysis. The punched tissue samples are enriched for the targeted brain regions; however, small amounts of surrounding tissue may be present.

**Extraction of polar metabolites from mouse BLA, dHIP and vHIP tissue samples and HILIC-MS for profiling of metabolites**. Polar metabolites from mouse (BLA, dHIP, and vHIP tissue samples; female: NS $n = 10$, LBN $n = 10$ & male: NS $n = 11$, LBN $n = 11$) were extracted and the untargeted analysis was performed using a Nexera UHPLC system (Shimadzu, Duisburg, Germany) coupled to a Q-TOF mass spectrometer (TripleTOF 6600 AB Sciex, Darmstadt, Germany). The untargeted metabolomics data has been deposited at MassIVE (https://massive.ucsd.edu/) under the number MSV000096480. See Supplemental Methods for details.

## Statistics and reproducibility

Statistical analyses and graphs were made using RStudio (with R 4.1.1), except for the unsupervised DeepOF analysis, which was performed using Python (v 3.9.13). Data were tested for the corresponding statistical assumptions, which included the Shapiro–Wilk test for normality and Levene's test for heteroscedasticity. If assumptions were violated the data were analyzed using non-parametric variants of the test. The group comparisons were analyzed using the independent samples t-test (T) as a parametric option, Welch's test (We), if data was normalized but heteroscedastic, or the Wilcoxon test (Wx) as a non-parametric option. Time-binned data was analyzed using the two-way repeated measures ANOVA with the phase as a within-subject factor and the condition as a between-subject factor. Data that showed a significant main effect were further analyzed with the post-hoc Bonferroni test (parametric) or the Kruskal-Wallis test (non-parametric). P-values were adjusted for multiple testing using the Bonferroni method. Timeline and bar graphs are presented as mean ± standard error of the mean (SEM). Data were considered significant at $p < 0.05$ (*), and further significance was represented as $p < 0.01$ (**), $p < 0.001$ (***), and $p < 0.0001$ (****). The description of sample size and cohort of animals is explained in every corresponding figure legend. Numerical source data for all graphs in the manuscript can be found in Supplementary Data 1.

## Reporting summary

Further information on research design is available in the Nature Portfolio Reporting Summary linked to this article.

## Results

### The physiological hallmarks of limited bedding and nesting (LBN) exposure

Chronic stress during early life was induced to assess the sex-dependent stress effects directly after early stress exposure and in adult age (Fig. 1A). A common consequence of LBN exposure is the reduction in body weight at P09[50]. The current study observed a significant reduced body weight in males and females, demonstrating a successful induction of chronic stress exposure during early life (Fig. 1B). Interestingly, the stress-induced reduction in body weight was sustained in female adult age, but not in males (Fig. 1C). A marker for chronic stress exposure and dysregulation of the HPA axis is the relative weight of the adrenals, which in adult age was not altered in females, but was significantly elevated in LBN males (Fig. 1D). In addition, as a proxy for stress exposure, the CORT levels were obtained directly after the LBN exposure at P09, the female LBN animals showed a

significant elevation of CORT levels, whereas in males no elevated CORT levels were observed (Fig. 1E). In adult age, no difference in basal CORT levels was observed for stress exposure in both sexes (Fig. 1F).

### LBN increased *FKBP5* expression in the dorsal hippocampus exclusively in males

*FKBP5* mRNA expression in the BLA (Supplementary Fig. 1A) and dorsal HIP, separating the subregions; CA1, CA2-3, and DG (Supplementary Fig. 1B) was assessed directly after LBN exposure at P09 and in adult age. Female and male data at P09 did not show a stress-induced difference of *FKBP5* expression in the BLA, CA1, CA2-3, and DG (Supplementary Fig. 1C, D), however, an indication of elevated *FKBP5* expression in LBN animals could be observed in males, but this was not significant (Supplementary Fig. 1D). Moreover, also during adult age, the female data did not indicate any *FKBP5* expression differences between stress conditions (Supplementary Fig. 1E). However, adult males showed a significant stress-induced increase of *FKBP5* expression in the CA1 region, but not in the BLA, CA2-3, and DG regions (Supplementary Fig. 1F). Moreover, both NS and LBN mice showed an age-dependent expression pattern of *FKBP5* regardless of sex in the dorsal HIP. At P09 *FKBP5* expression was the highest in the CA2-3 and showed similar levels of expression in the CA1 and DG, whereas at p56, the *FKBP5* expression was the highest in the DG, followed by the CA2-3, and the CA1 (Supplementary Fig. 1C–F).

### Freezing behavior is affected by LBN exposure in a sex-specific manner

During the acquisition of fear conditioning, the typical increase of freezing behavior over the different tone representations was observed in both females and males, regardless of the stress condition (Fig. 2A–D). However, an overall decrease during the fear acquisition in freezing behavior was observed at the tone representations in males, but not females (Fig. 2B, D). Moreover, the exploration during the ITIs of the fear acquisition phase showed that the freezing at the individual ITIs was not significantly altered in females based on the stress condition (Fig. 2E), but there was a significant reduction in the freezing response of LBN females in the overall mean ITIs (Fig. 2F), which was not observed in males (Fig. 2G, H). When looking at the recall of fear, the mean freezing during the contextual fear memory was significantly lowered in LBN females and males (Fig. 2I, Supplementary Fig. 2A, B). The auditory fear retrieval did not show a different freezing response on mean tones in females (Fig. 2J, Supplementary Fig. 2C), but did show a lowered freezing response in LBN males compared to NS (Fig. 2J, Supplementary Fig. 2D). In addition, no differences were observed in the fear retrieval for both females and males (Fig. 2K, Supplementary Fig. 2E, F).

Next, the DeepOF unsupervised clustering analysis of behavioral data was employed to capture a broader range of behavioral differences beyond just freezing behavior. The DeepOF unsupervised clustering analysis using the DeepLabCut pose-estimation of the male and female fear acquisition tone data resulted in an optimum number of 9 distinct clusters, which could potentially point to clusters of a specific behavioral type (Fig. 3A, B; clusters 0–8). No cluster population differences were observed by the stress background for both female and male ITI 1–4 data, indicating that no additional behavioral clusters were altered (Supplementary Fig. 3A, B). However, two clusters were significantly altered by the stress background in females during tones 2–5 (Fig. 3A), which was not observed in males (Fig. 3B). In particular, "cluster 0" was significantly increased in LBN females, and "cluster 6" significantly decreased in LBN females compared to NS (Fig. 3A), indicating that these clusters of behavior are relevant for the analysis of sex and ELS exposure. A multi-class supervised learning model was trained to map motion summary statistics to the obtained cluster labels, and performance was measured in terms of the balanced accuracy per cluster, showing a performance of at least 0.6 or higher (Fig. 3C). In addition, the confusion matrix showed low probabilities for all cluster crossovers, and the classifier performance was substantially greater than random for all clusters, indicating that all clusters were substantially distinguishable by the model (Fig. 3D). The cluster detection analysis yields a set of feature explainers per

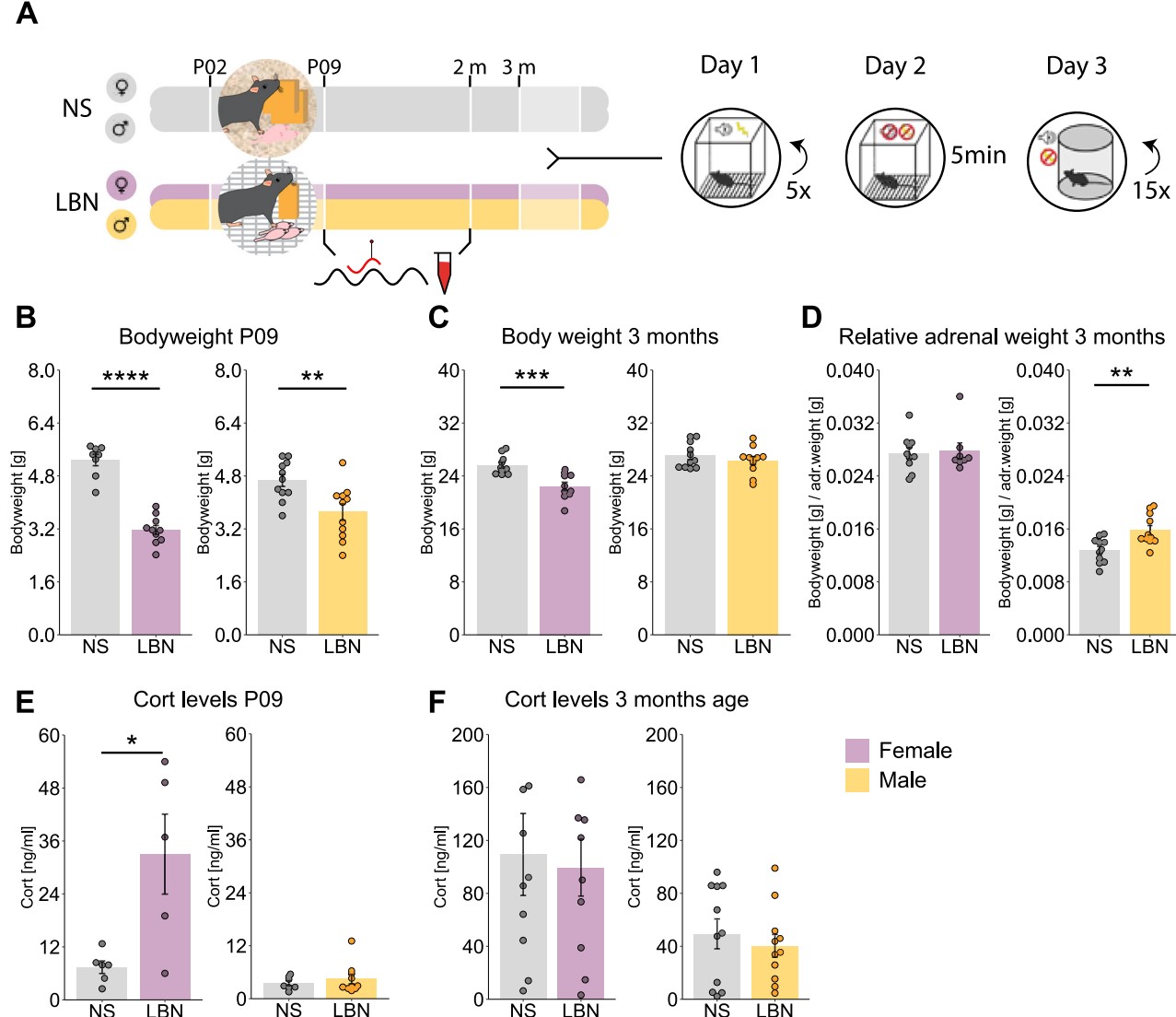

**Fig. 1 | Physiological stress hallmarks of limited bedding and nesting (LBN).**
**A** Experimental timeline for non-stressed (NS) controls and limited bedding and
nesting (LBN) stressed mice on fear conditioning. **B** Significant decrease in body
weight was observed after LBN exposure at P09 for females (Wx = 80, $p < 0.0001$),
and males (T(20) = 3.13, $p = 0.005$). **C** During adult age (3 months) the body weight
was significantly reduced in females (T(18) = 4.16, $p = 0.0006$), but not in males
(T(20) = 0.88, $p = 0.39$). **D** The relative adrenal weight was not altered in females
(Wx = 45, $p = 0.70$), but was significantly increased in LBN males (T(20) = −3.3,
$p = 0.003$). **E** At P09, CORT levels were significantly elevated in LBN females

(Wx = 4, $p = 0.05$), whereas this was not the case for males (Wx = 31, $p = 0.74$). **F** At
adult age, the CORT levels were not altered by stress for both females (T(18) = 0.26,
$p = 0.79$) and males (T(20) = 0.64, $p = 0.53$). The bar graphs are presented as
mean ± standard error of the mean and all individual samples as points. Panels B-D,
F represent female NS ($n = 10$), female LBN ($n = 10$), male NS ($n = 11$), male LBN
($n = 11$). Panel E represent female NS ($n = 6$), female LBN ($n = 5$), male NS ($n = 7$),
male LBN ($n = 10$).

cluster that can be used to interpret the clusters using SHAP values in global
(Fig. 3E) and cluster-specific ways (Fig. 3F, G), which enables the explora-
tion of the type of behavior that is represented in an unsupervised cluster.
Importantly, the interpretation of the clusters was performed using the
feature importance of the SHAP analysis, together with the visual inter-
pretation of the video fragments per cluster (see supplemental materials for
video output per cluster). The global feature importance across all clusters
revealed that the distance towards several spine labels (a stretch or a
shortening of the back), the overall speed (an increased or reduced speed),
huddle (an increased or decreased amount of the behavior in which the
animal stops moving around and bends the back), and the surface area of the
head (an increased surface area is related to the head being forward, whereas
a decreased surface area is related to the head being downward) were par-
ticularly important for global cluster inclusion (Fig. 3E). More specifically,
the feature importance analysis for "cluster 0" revealed that an increased

speed, a decrease in huddle behavior, and an increased spine stretch were
important features for cluster inclusion (Fig. 3F). The visual inspection of
"cluster 0" indicated a behavior related to the exploration of the environ-
ment, in which the animal was moving around (see Supplemental Videos). In
contrast, "cluster 6" feature importance analysis revealed that a decrease
in spine stretch, an increase in huddle behavior, and a reduction of the
surface head area were important features for cluster inclusion (Fig. 3G).
The visual inspection of "cluster 6" indicated a behavior related to freezing,
in which the animal was often immobile and close to the outside of the
environment.

**Sex-dependent LBN exposure affect the expression of metabolic
pathways in the BLA**
In order to determine the intertwined relation between sex-specific and
brain region-specific stress effects with global metabolic cascades, a

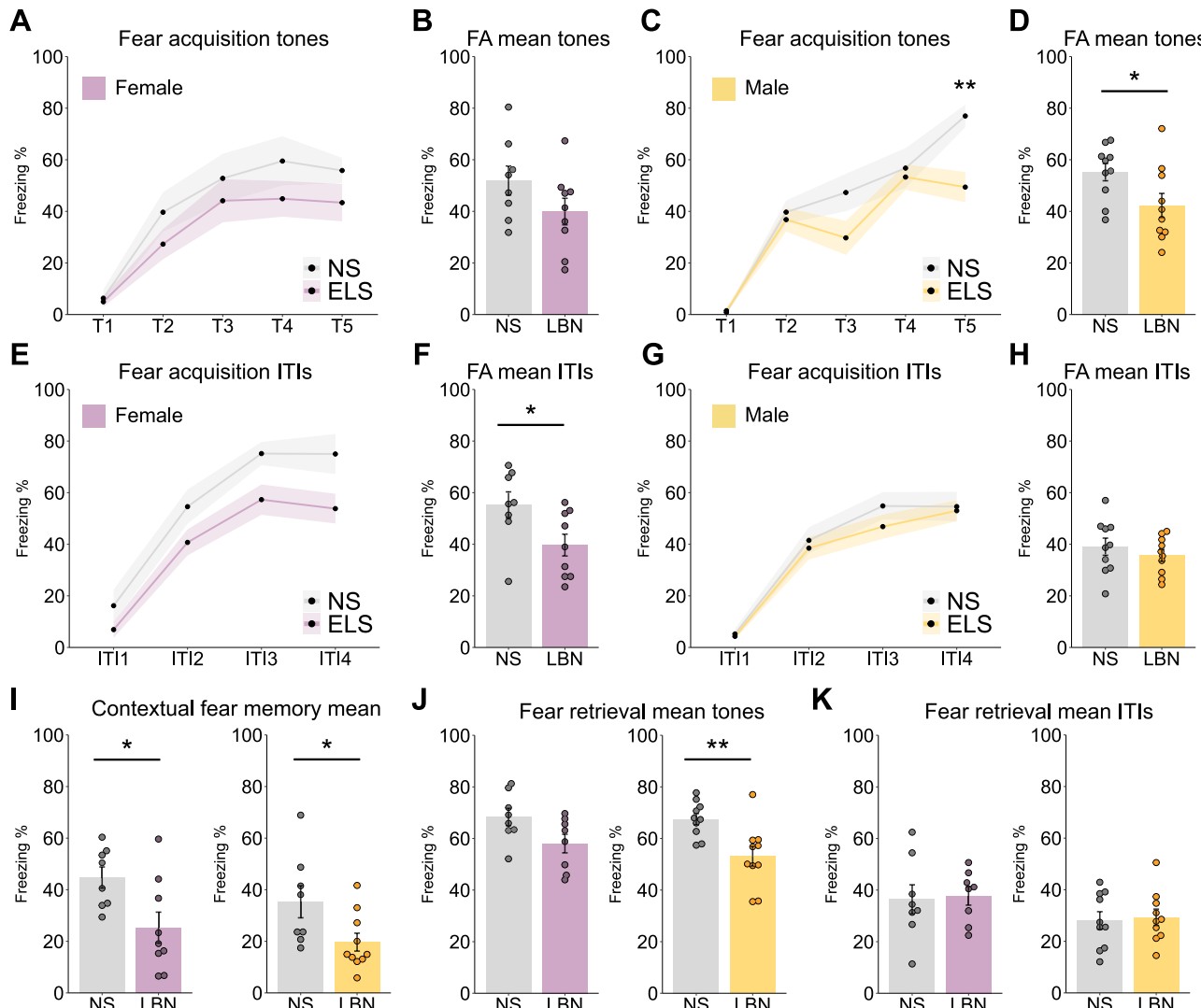

**Fig. 2 | Fear conditioning data on freezing behavior. A** Freezing behavior on individual tones during fear acquisition (FA) was not altered between non-stressed (NS) and stressed (LBN) females, two-way ANOVA ($p > 0.45$). **B** The average freezing during the FA tones 2–5 was not altered between NS and LBN females ($T(15) = 1.58$, $p = 0.14$). **C** Freezing behavior in the FA for individual tones 1–4 was not altered between NS and LBN males ($p > 0.22$), but for tone 5 it was significantly lowered in LBN males ($F(1,18) = 14.3$, $p = 0.005$), with a significant main effect for the two-way ANOVA on stress ($F(1,90) = 9.39$, $p = 0.003$), tones ($F(4,90) = 41.30$, $p < 0.0001$), and stress*tones ($F(4,90) = 2.6$, $p = 0.041$). **D** The average freezing during FA tones 2–5 was significantly lowered in LBN males compared to NS ($T(18) = 2.25$, $p = 0.037$). **E** No significant main effect could be observed between NS and LBN females for the individual ITIs in the FA. **F** The average freezing during FA ITIs 1–4 was significantly lowered in LBN females compared to NS ($T(15) = 2.37$,

$p = 0.03$). **G** No significant main effect could be observed between NS and LBN males for the individual ITIs in the FA. **H** The average freezing during FA ITIs 1–4 was not altered between NS and LBN males ($T(18) = 0.84$, $p = 0.41$). **I** The mean freezing in the contextual fear memory task was significantly lowered in LBN females compared to NS ($T(15) = 2.61$, $p = 0.020$), and in males ($T(16) = 2.32$, $p = 0.034$). **J** The mean freezing in the fear retrieval task showed no significant difference in females ($T(14) = 2.09$, $p = 0.055$). However, in males, a significant reduction in freezing was observed in LBN compared to NS ($T(18) = 3.29$, $p = 0.004$). **K** No significant differences were observed in the mean freezing during the ITIs between LBN and NS animals for females and males. The timelines and bar graphs are presented as mean ± standard error of the mean and all individual samples as points. Panel **A–K** represent female NS ($n = 10$, female LBN ($n = 10$), male NS ($n = 10$), male LBN ($n = 11$).

comprehensive and highly sensitive metabolomic measurement was conducted in the BLA, dHIP and vHIP. The metabolic analysis revealed a significant regulation (down or up) in all brain regions for sex, by comparing males and females regardless of stress exposure in HILIC negative mode (BLA: 62, dHIP: 54, vHIP: 58 significant features, $p < 0.05$) and positive mode (BLA: 105, dHIP: 107, vHIP: 120 significant features) (Supplementary Fig. 4A–C). The regulation through stress did not reveal a pronounced regulation of metabolic cascades in all brain regions, comparing NS and LBN conditions, regardless of sex in HILIC negative mode (BLA: 0, dHIP: 1, vHIP: 1 significant features) and positive mode (BLA: 2, dHIP: 3, vHIP: 9 significant features) (Supplementary Fig. 4D–F).

Next, further analysis investigated the metabolic regulation on sex between males and females and were subsequently compared between conditions, NS and LBN exposure. A significant regulation of metabolites in the sex comparison (male vs female) was observed in all brain regions for the NS condition HILIC negative mode (BLA: 49 significant features; corresponding to 32 significant validated metabolites, dHIP: 42 significant features; corresponding to 29 significant validated metabolites, vHIP: 50 significant features; corresponding to 24 significant validated metabolites) and HILIC positive mode (BLA: 85 significant features; corresponding to 46 significant validated metabolites, dHIP: 91 significant features; corresponding to 39 significant validated metabolites, vHIP: 52 significant features; corresponding to 33 significant validated metabolites). In addition,

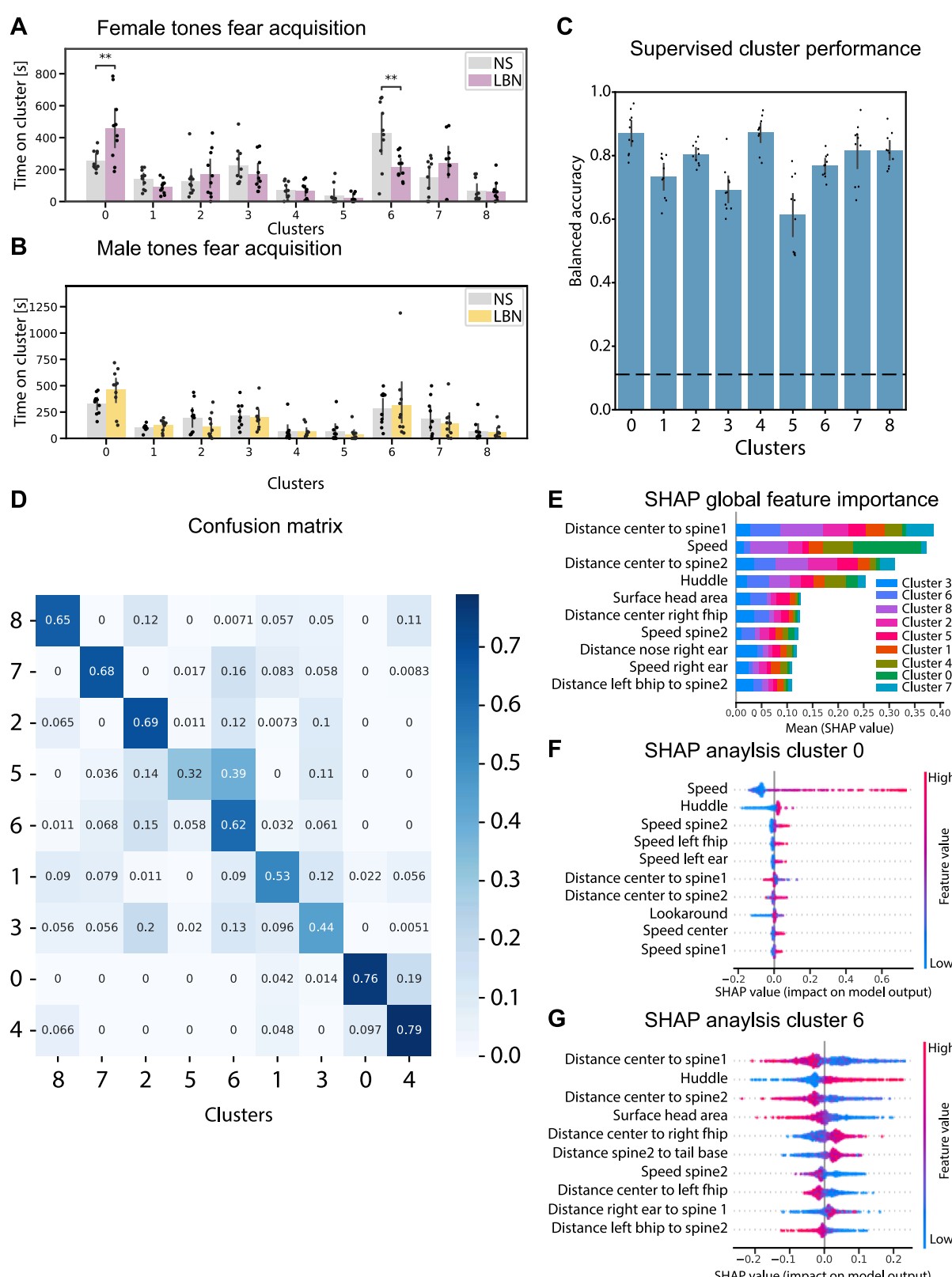

the regulation of metabolites comparing sex (male vs female) in the LBN condition also showed a significant regulation in HILIC negative mode (BLA: 51 significant features; corresponding to 33 significant validated metabolites, dHIP: 39 significant features; corresponding to 28 significant validated metabolites, vHIP: 52 significant features; corresponding to 33 significant validated metabolites) and HILIC positive mode (BLA: 81 significant features; corresponding to 41 significant validated metabolites, dHIP: 89 significant features; corresponding to 37 significant validated metabolites, vHIP: 103 significant features; corresponding to 42 significant validated metabolites) (Supplementary Fig. 5A–C). The lists of altered metabolites across the different brain regions are provided in Supplementary Data 2.

**Fig. 3 | Unsupervised clustering analysis of fear acquisition data on tones 2–5. A** Cluster enrichment for the female fear acquisition data using tones 2–5. Bar graphs represent mean ± standard deviation of the time proportion spent on each cluster. Statistics were performed using an independent samples t-test corrected for multiple testing using Benjamini–Hochberg's method across clusters between NS and LBN exposure. Significant differences were observed in clusters 0: T = −3.55, $p = 2.28^{e-3}$, and 6: T = 4.24, $p = 4.97^{e-4}$, but none of the other clusters ($p > 0.05$). **B** Cluster enrichment for the male fear acquisition data using tones 2–5. No significant differences were observed between NS and LBN mice using the independent samples t-test corrected for multiple testing using Benjamini–Hochberg's method across clusters ($p > 0.05$). **C** The supervised cluster performance was performed via the validation performance per cluster across a 10-fold cross-validation loop. Balanced accuracy was used to correct for cluster assignment imbalance. The dashed line marks are the expected performance due to chance, considering all outputs. **D** The confusion matrix is obtained from the trained gradient boosting machine classifying between clusters. Aggregated performance over the validation folds of 10-fold cross-validation is shown. **E** The global SHAP feature importance between the different clusters. Features in the y-axis are sorted on the global absolute SHAP values across all clusters. The classes in the bar graphs are sorted by highest to lowest clusters importance within every feature. **F, G** Bee swarm plots for the two differentially expressed clusters within the female fear acquisition data between NS and LBN mice, clusters 0 and 6. The plots show the 10 most important features for each classifier, in terms of the mean absolute value of the SHAP values. Bar graphs represent mean ± standard deviation of the time proportion spent on each cluster. Panel **A–F** represent female NS (n = 10, female LBN (n = 10), male NS (n = 10), male LBN (n = 11).

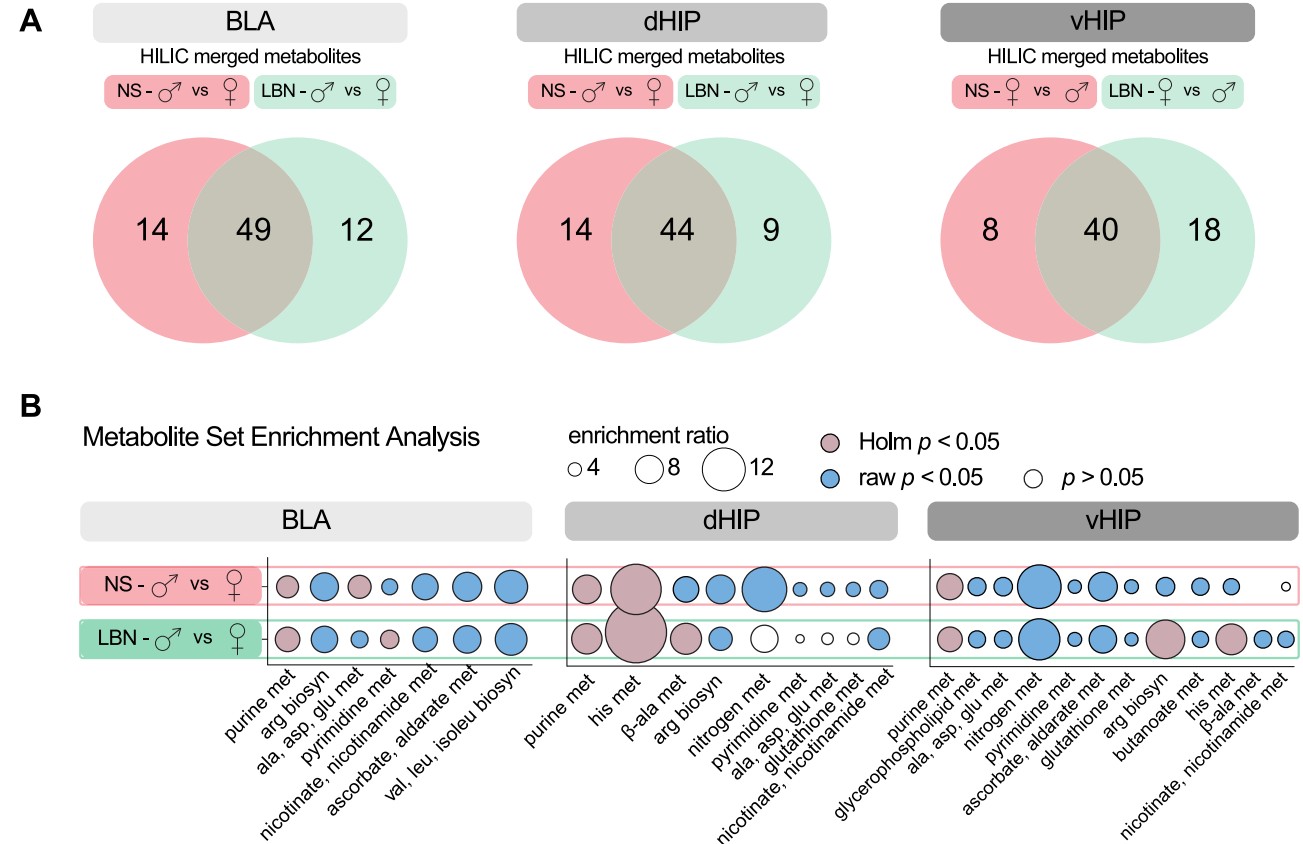

**Fig. 4 | Metabolomics reveals altered glutamate, pyrimidine, and purine pathways related to sex and stress condition. A** The set of regulated metabolites in the male vs female comparison between NS and LBN conditions, with a dedication amount of specific regulated metabolites in only NS or only LBN conditions: BLA: 34.67%, 26 out of 75 metabolites, dHIP: 34.33%, 23 out of 67 metabolites, vHIP: 39.39%, 26 out of 66 metabolites. **B** Metabolite Set Enrichment Analysis (MSEA) revealed different pathways of interest in the BLA, dHIP, and vHIP. 1) purine metabolism was significantly regulated in **NS** (BLA: Holm test; $p = 1.28^{e-6}$; enrichment ratio (ER) = 7.32, dHIP: Holm test; $p = 1.21^{e-10}$; ER = 9.68, vHIP: Holm test; $p = 3.8^{e-9}$; ER = 8.81) and **LBN** (BLA: Holm test; $p = 1.43^{e-8}$; ER = 8.09, dHIP: Holm test; $p = 2.02^{e-9}$, ER = 10.24, vHIP: Holm test; $p = 9.33^{e-9}$, ER = 8.33). 2) pyrimidine metabolism was significantly regulated in LBN in the BLA (Holm test; $p = 0.021$; ER = 6.23), but not in NS (Holm test; $p = 0.13$; raw p-value = 0.0017; ER = 5.48), but not in the dHIP and vHIP. 3) glutamate metabolism; the pathway "alanine, aspartate and glutamate metabolism" was significantly regulated only in the BLA in NS (Holm test; $p = 0.028$; ER = 7.63), but not in LBN (Holm test; $p = 0.33$; raw p-value = 0.0042; ER = 5.79). The "arginine biosynthesis" was significantly regulated only in the vHIP in LBN (Holm test; $p = 0.019$, ER = 11.87), but not in NS (Holm test; $p = 1$, raw p-value = 0.039, ER = 6.23). Panels **A** and **B** represent female NS (n = 10, female LBN (n = 10), male NS (n = 10), male LBN (n = 11).

The different HILIC settings (negative and positive mode) can reveal partially different metabolites and were merged to obtain a complete list of significantly regulated metabolites (BLA: NS: 63 and LBN: 61, dHIP: NS: 58 and LBN: 53, vHIP: NS: 48 and LBN: 58 merged and significant validated metabolites). The set of regulated metabolites in the male vs female comparison between NS and LBN conditions showed that a proportion of the metabolites were specifically regulated in only NS or only LBN conditions (BLA: 34.67%, 26 out of 75 metabolites, dHIP: 34.33%, 23 out of 67 metabolites, vHIP: 39.39%, 26 out of 66 metabolites) (Fig. 4A). These results point to the fact that there might be specific regulated pathways underlying these differentially regulated metabolites between stress conditions while comparing male vs female.

Next, a Metabolite Set Enrichment Analysis (MSEA) was performed, which aims to understand the biological significance of the enriched sets of

metabolites, associated with specific biological pathways or processes between sexes comparing NS and LBN conditions. The MSEA revealed several pathways of interest when comparing between the NS and LBN conditions in relation to stress; 1) the purine metabolism pathway, 2) the pyrimidine metabolism pathway, and 3) the glutamate metabolism pathways, as observed in the "arginine biosynthesis" and "alanine, aspartate and glutamate metabolism" pathways (Fig. 4B).

The purine metabolism pathway showed a significant regulation in NS male vs female, as well as in LBN male vs female in all brain regions (Fig. 4B). In addition, the comparison NS and LBN showed similar enrichment ratios indicating a differential regulation of the purine metabolism pathway between males and females, regardless of stress exposure. The pyrimidine metabolism pathway showed no regulation in NS male vs female, but did show a significant regulation in LBN male vs female, while the enrichment ratios were similar between conditions, which was exclusively observed in the BLA (Fig. 4B). The glutamate metabolism pathway, via the "alanine, aspartate and glutamate metabolism", but not in "arginine biosynthesis" showed a significant regulation in NS male vs female, but not LBN male vs female exclusively the BLA (Fig. 4B). Interestingly, an opposite effect was observed in the vHIP, which showed a significant regulation in LBN male vs female, but not NS male vs female for "arginine biosynthesis", but not in "alanine, aspartate and glutamate metabolism". No further differences were observed in the dHIP for the glutamate metabolism pathway (Fig. 4B). These results indicate the differential regulation of the glutamate metabolism pathway between NS and LBN when comparing males and females.

## Discussion
Exposure to ELS increases vulnerability to stress-related disorders, such as PTSD. The prevalence of PTSD is strongly influenced by sex. Animal models investigating ELS, in particular by using the limited bedding and nesting (LBN) model, have provided increasing evidence that LBN exposure affects rodents in a sex-specific manner[39–43]. However, the sex-dependent impacts of HPA-axis signaling, metabolomic pathway analysis in stress-related brain regions, and fear memory formation on ELS remain to be fully elucidated. The development of open-source markerless pose estimation tools[57] and subsequently unsupervised behavioral analysis tools has enabled in-depth analysis that can explore previously unknown behavioral patterns[58]. This advancement is crucial for enhancing the understanding of the behavioral outcomes associated with stress-induced fear memory formation. The present study contributes to a better understanding of those mechanisms by investigating HPA axis signaling in the body and brain, along with metabolomic pathway alterations in the BLA and behavioral responses to LBN-induced fear memory formation in a sex- and time-dependent manner.

### LBN disrupts different facets of the HPA axis in a time- and sex-dependent manner
The LBN model has been extensively utilized to investigate the effects of chronic ELS exposure on both physiological and behavioral outcomes. A common hallmark of LBN exposure is the reduction of body weight at P09[50], which was confirmed in the current study, highlighting that the LBN model has stress-dependent effects in both sexes[59,60]. In addition, the long-term effect of LBN on body weight between sexes is more variable and seems to be dependent on the age of testing and potential additional challenges throughout adulthood. The current study showed that at 3 months of age, females show a more persistent LBN-related body weight reduction phenotype compared to males. Other studies shown a similar effect at two months of age[59], but Arp et al. did not find this sex-dependent difference in 4 months old animals[61], indicating that both males and females eventually recover, but females show a longer recovery period. An opposite effect was observed in another study at eight months of age, where males showed a stronger body weight reduction compared to females[60], this might be explained by the different adult stress events (e.g., glucose- and insulin tolerance tests) and indicates different vulnerability toward such adult stressors after LBN exposure between sexes.

ELS exposure has been linked to dysregulation of the HPA axis, which can lead to an increased vulnerability state of stress-related psychopathology[62,63]. A well-established phenomenon is the elevated levels of morning CORT baseline in females compared to males[64–66]. The current study replicates this phenomenon in adult mice regardless of stress exposure, as also recently reported by Brix et al.[60]. Interestingly, it was further observed that this sex-dependent difference is already apparent at the early age of P09, highlighting that the sex-specific differences are apparent already at the end of ELS exposure. An earlier study on LBN exposure in males showed a significant LBN-induced increase at P09 for baseline CORT in mice[50], but the current study found an increase in CORT only in LBN females, albeit a high variability in the LBN group. This might be explained due to the low baseline levels of CORT in males, which therefore, might show a higher variance, as the absolute CORT values between conditions are smaller. Nonetheless, the high increase of baseline CORT in LBN-exposed females is a good proxy for stress exposure, and at P09 is indicated to be higher in females compared to males. It is intriguing to speculate that females with a particularly large CORT response to LBN would later also be more affected on behavioral and molecular alterations. Moreover, we observed an opposite effect for adrenal weight at adult age, which was significantly increased for LBN-exposed males, but not females. This is in line with earlier research, that showed a similar effect in males at 1 month of age[61], but at later stages in adulthood, namely 4 months and 8 months, the adrenals in males were back to the same size as the non-stressed condition[50,60]. This indicates that the adrenal size is influenced in a time-dependent manner, in which males are taking longer to recover their adrenal size to baseline after LBN exposure. Another facet of the HPA-axis reactivity was investigated via gene expression changes of *FKBP5* in the brain. Previous research has identified a particularly high expression of the *FKBP5* gene in the BLA and HIP under baseline, which is confirmed in the current study[35]. Furthermore, *FKBP5* gene expression has been shown to increase in a stressor-dependent manner in the BLA and HIP[35,67,68], but the immediate and long-term *FKBP5* gene expression changes in response to LBN have remained elusive. We show that *FKBP5* gene expression was not changed by LBN exposure directly after the stress at P09 in both sexes but was upregulated specifically in the CA1 region of the dorsal HIP of adult LBN-exposed males, but not females. Marrero et al. 2019[36] showed that the overexpression of human *FKBP5* in the forebrain induces specific downstream molecular changes in the dorsal HIP in adult ELS animals using the maternal separation paradigm. This coincides with the current finding that specifically the dorsal HIP shows upregulated *FKBP5* expression and points to an altered molecular pathway mechanism after LBN exposure. In conclusion, we highlight a differential impact of LBN exposure across sexes. The immediate effects of LBN exposure at P09 are more pronounced in females, while interestingly, the prolonged effects of a dysregulated HPA-axis in adult age are affected exclusively in males.

### Fear acquisition is differentially affected by LBN across sexes
The formation of fear memory is a crucial aspect of understanding the underlying mechanism of PTSD. The specific alterations of anxiety and fear behavior can be investigated using animal models of fear conditioning. In line with previous research, we show that exposure to LBN reduces the fear response by lowering freezing behavior during both contextual- and auditory fear retrieval in males[46,47]. In addition, we show that LBN exposure in females shows a similar reduction in freezing behavior during the contextual fear retrieval, but not in the auditory fear retrieval. Previous research has shown that LBN is linked to reduced synaptic plasticity markers within the dorsal HIP, which could explain the LBN-induced differences in fear retrieval by altering fear memory formation[46]. However, the current study highlights an alternative explanation by specifically exploring the fear behavior during the acquisition of fear conditioning. Specifically, it was observed that the reduction in freezing behavior can already be found during the acquisition of the fear memory, in which LBN-exposed males show reduced freezing during the acquisition tones, and the females during the acquisition ITIs. A similar effect has been observed in other studies for male

data, in which it was shown that the freezing directly after the fear acquisition is already lowered in LBN-exposed animals[46,61,69]. Therefore, the difference in freezing during the retrieval phase is not only explained by differences in fear memory formation but also by an altered response at fear acquisition.

In addition, several studies have highlighted the relevance of distinguishing between different types of fear behaviors[70–72]. The analysis of the behavioral data, using an unsupervised analysis, provides a promising way to explore novel behavioral patterns related to fear acquisition without prior behavioral categorization. This allows the exploration of the behavioral repertoire in a hypothesis-generating way, which can lead to the identification of novel behaviors within the specific methodological context[73]. To further understand the sex-dependent fear behavioral differences during fear acquisition, the DeepOF open-source python package was deployed to perform an unsupervised analysis pipeline, which maps the representations of different fear behavior-related syllables across the different stages (tones and ITIs), conditions (NS vs LBN) and sex (female vs male) without any prior label information. Different fear-related behaviors were observed that were particularly altered in female, but not male mice. Interestingly, it was observed that specific behavioral clusters (e.g., cluster 0) are elevated in LBN-exposed females, which indicated a behavior related to the exploration of the environment. Conversely, other behavioral clusters (e.g., cluster 6) are reduced in LBN-exposed females, which were related to freezing behavior. Intriguingly, the observed behavior in "cluster 0" coincides with a previously identified active fear behavioral response, called "darting", in which rapid locomotive movements are detected in primarily female rats[72]. The behavioral syllables from "cluster 0" can also be allocated to an active fear behavioral response, but under non-stressed conditions are expressed in both females as well as males. However, the increased amount of the expression of "cluster 0" after LBN exposure is exclusively observed in females, which does indicate a sex-dependent effect on the active fear response.

Of note, LBN exposure is known to impact a wide range of behavioral domains, including social dominance[74], cognition[75,76], and anxiety[77]. These behavioral changes can indeed influence fear memory formation measured during the fear conditioning protocol, and conversely, alterations in fear memory could affect these other behavioral outcomes. Further research is needed to disentangle the contributions of these various neural circuits and determine potential overlaps among the different behavioral changes induced by LBN.

### Brain metabolomics reveals sex and stress dependent alterations in purine, pyrimidine, and glutamate metabolism pathways

The exploration of metabolic mechanisms and pathways in the brain in relation to sex and LBN exposure has remained unexplored. The current study highlights the importance of investigating cellular metabolism and the subsequent downstream analysis of metabolic pathways after LBN exposure in the BLA, dHIP and vHIP to elevate the understanding of the neurobiological etiology of stress-related disorders. Interestingly, a similar pattern of metabolic regulation was observed in the BLA, dHIP, and vHIP, where a significantly high number of metabolites differed between sexes but not in response to LBN exposure. This was further highlighted by MSEA pathway analysis, which revealed altered purine metabolism between the sexes in all brain regions. Previous studies have reported substantial differences in metabolic regulation between sexes in brain tissue, primarily in the context of disorders such as obesity[78] or Alzheimer's disease[79,80], but not under naive conditions. Moreover, the purine metabolism pathway has previously been linked to sex-specific differences that influence vulnerability to metabolic diseases, as observed in blood plasma[81]. The current study reveals that purine metabolism is not only altered peripherally between sexes but also in several brain regions, including the BLA, dHIP, and vHIP. These findings highlight, for the first time, a significant role for metabolic regulation in the BLA, dHIP, and vHIP in contributing to sex-dependent differences between males and females. This underscores the importance of sex-segregated analysis in studies of stress exposure. In addition, the differential metabolic

profiles between sexes could also contribute to the explanation of the differential behavioral responses after LBN exposure. This was highlighted in particular in the BLA, which showed a significantly altered regulation between males and females in the NS condition, but not LBN condition in the "alanine, aspartate and glutamate metabolism" pathway. A possible explanation could be that LBN exposure alters glutamate metabolism in a similar way in males and females, while a strong differential effect can be observed under baseline conditions between sexes. Intriguingly, a recent study showed that there is a disbalance in GABAergic and glutamatergic gene expression levels in the ventral hippocampal neurons after LBN exposure in male mice[74]. Taking into account the strong connection between the HIP and the BLA after stress[82–84] and their similar expression of stress-responsive genes, such as GR, MR and *FKBP5*, it is possible that a similar disbalance could be observed in the BLA. This would suggest that beyond the differences in gene transcription, the current study shows that pathways related to glutamate metabolism in the BLA are altered as well in a sex-dependent manner for LBN exposure. In addition, Kos et al.[74] show that changing the brain-wide balance between glutamatergic vs GABAergic signaling in LBN-exposed mice restores the ELS-induced affected behaviors, highlighting the importance of the balance between glutamate and GABA in ELS affectedness. Considering the current study, it is possible that this rescue effect is mediated partially via restoring metabolic processes in the BLA. One limitation of the current approach is that all animals were exposed to the fear conditioning paradigm, so it is possible that differences in the brain metabolomic profile could only be unmasked following a severe adult stress exposure, while not being detectable under baseline conditions.

## Conclusion

Taken together, the current study shows a sex-specific effect of LBN exposure on dysregulation of the HPA-axis, in which the adrenal weight, baseline CORT levels are altered in a time-dependent manner. In addition, we show that specific aspects of fear-related behavior, including the passive fear behavioral response via freezing behavior, but also an active fear response, as identified using an unsupervised analysis, are altered by LBN exposure in a sex-specific manner. The additional fear-related behavior that is expressed in "cluster 0" is contributing to a better understanding of the sex-dependent effects of fear memory acquisition and might influence the expression of the freezing behavior during contextual as well as auditory fear retrieval. The DeepOF unsupervised analysis provides an additional layer to explore the fear-related behaviors without prior assumptions and therefore, allows for hypothesis-generating behavioral analysis, which ultimately can lead to a better understanding of the stress-induced behavioral phenotype. Additionally, this research is the first to discover sex-related differences in purine metabolism within the BLA, dHIP, and vHIP brain regions, and it reveals that the interplay between sex and early-life stress influences glutamate metabolism, particularly in the BLA. The current study proposes a potential link between these metabolic alterations and the observed fear-related behavioral outcomes, emphasizing the need for a comprehensive understanding of the metabolomic mechanisms underlying LBN exposure. Overall, these findings highlight the importance of considering sex-specific metabolic responses in understanding the neurobiological mechanisms underlying stress-related disorders and offer potential avenues for targeted interventions.

## Data availability

All data and accompanying scripts for the DeepOF unsupervised analysis are deposited at Zenodo (https://zenodo.org/records/14237774)[85]. The functionality of the DeepOF unsupervised analysis can be found on the read the docs website (https://deepof.readthedocs.io). The untargeted metabolomics data has been deposited at MassIVE (https://massive.ucsd.edu/)[86] under the number MSV000096480. Numerical source data for all graphs in the manuscript can be found in Supplementary Data 1.

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

## Acknowledgements
The authors thank Lisa Rudolph, Daniela Harbich, and Bianca Schmid for their technical assistance, and the DeepLabCut development team for creating and maintaining the DeepLabCut software. M.V.S. is funded by the "Kids2Health" grant of the Federal Ministry of Education and Research [01GL1743C]. M.V.S. and N.C.G. are funded by Deutsche Forschungsgemeinschaft (DFG, German Research Foundation, 453645443). L.M. is funded through the European Union's Horizon 2020 research and innovation program under the Marie Skłodowska Curie grant agreement 813533.

## Author contributions
J.B. and M.V.S. conceived the study. J.B. performed the experiments, L.v.D., L.M.B., S.N., H.Y., S.M., V.K. and M.S. assisted with the experiments. Metabolomics was performed by K.K. and analyzed by T.B. and N.G. All other data was analyzed by J.B. and with the assistance of L.M. and B.M.M. J.B. wrote the first draft of the manuscript and all authors contributed to the final manuscript version.

## Funding

## Competing interests
The authors declare no competing interests.
