## [Transparent Peer Review file · Communications Biology]

Sex-specific fear acquisition following early life stress is linked to amygdala and hippocampal purine and glutamate metabolism

Corresponding Author: Dr Mathias Schmidt

This manuscript has been previously reviewed at another journal. This document only contains information relating to versions considered at Communications Biology.

Version 0:

Reviewer comments:

Reviewer #2

(Remarks to the Author)

In this manuscript, Bordes and Coll aim to investigate the effects of gender on LBN-induced fear-related behaviors. Moreover, they also examined the impact of this experimental manipulation on cellular metabolism in critical brain structures in psychopathologies and stress.

Although these are correlational data, they are interesting insights, especially the deep behavioral analysis performed. The manuscript has already been reviewed by three experts, significantly improving the work compared to its first version. For this reason, I have no further changes and requests for the authors, and I consider the paper adequate for this journal recommending its publication in its current version.

Reviewer #3

(Remarks to the Author)

Overall, I find the findings of the authors very interesting and of relevance regarding the interactions between early life stress and stress-related disorders later in life. Moreover, animal research including females is necessary to advance our understanding on the above topic. In my view the authors have addressed the reviewer's comments. I have just a general remark/question. Did the authors try to correlate markers such as CORT or weight at P9 for example, with the behavioral readouts on the fear conditioning, (traditional definition of freezing and clusters) or the metabolomics-derived changes?

Below are minor comments:

Line: "However, the effects of fear memory formation on ELS have not been explored in detail", do the authors mean the opposite? i.e. the effects of ELS on fear memory formation?
232: typo? Should it be "represented"?
